# The Metabotropic Glutamate 5 Receptor in Sleep and Wakefulness: Focus on the Cortico-Thalamo-Cortical Oscillations

**DOI:** 10.3390/cells12131761

**Published:** 2023-06-30

**Authors:** Richard Teke Ngomba, Annika Lüttjohann, Aaron Dexter, Swagat Ray, Gilles van Luijtelaar

**Affiliations:** 1School of Pharmacy, University of Lincoln, Lincoln LN6 7DL, UK; 2Institute of Physiology I, University of Münster, 48149 Münster, Germany; 3Department of Life Sciences, School of Life and Environmental Sciences, University of Lincoln, Lincoln LN6 7DL, UK; 4Donders Centre for Cognition, Radboud University, 6525 XZ Nijmegen, The Netherlands

**Keywords:** sleep and wakefulness, metabotropic glutamate receptor 5, thalamocortical oscillations, epilepsy and sleep, mGlu5 receptor allosteric modulators

## Abstract

Sleep is an essential innate but complex behaviour which is ubiquitous in the animal kingdom. Our knowledge of the distinct neural circuit mechanisms that regulate sleep and wake states in the brain are, however, still limited. It is therefore important to understand how these circuits operate during health and disease. This review will highlight the function of mGlu5 receptors within the thalamocortical circuitry in physiological and pathological sleep states. We will also evaluate the potential of targeting mGlu5 receptors as a therapeutic strategy for sleep disorders that often co-occur with epileptic seizures.

## 1. Introduction

Sleep and wakefulness are interconnected but selectively activated by distinct neuronal circuits in the brain [1,2]. From a biological point of view, sleep is a universal complex state, with control mechanisms comprising gene regulation, intracellular processes, and neural network activities. It is characterised by an innate quiescent state which, behaviourally, can be observed across various species (ranging from mammals to invertebrates such as drosophila) [2,3]. This behaviour is comprised of two distinct states: rapid eye movement (REM) sleep and non-REM (NREM) sleep. Disturbances of both states have been linked to various metabolic disorders, including obesity and cardiovascular diseases, as well as to psychiatric and neurological disorders, such as Parkinson’s disease, Huntington’s disease, and Alzheimer’s disease [4,5]. Over the last 20 years, the structural organization of sleep-related brain activity in both space and time has, at least in part, been characterised. The hallmark of REM sleep, as seen in the electroencephalogram (EEG) or local field potentials (LFP), are the theta and the gamma rhythms, while in NREM sleep, slow delta wave oscillations, sleep spindles, and sharp wave ripples are the main constitutes of EEG and LFP [6]. These oscillations originate as circumscribed thalamocortical or hippocampal neural circuit activities, and the underlying cellular mechanisms are contingent on the intrinsic properties of various ion channels that are either regulated or modulated by glutamate and GABA receptors [6].

The importance of sleep is highlighted by the fact that sleep deprivation leads to an increased demand for sleep, as reflected in EEG recordings which show a compensatory increase in the power of delta oscillations during the NREM sleep stage. While the precise role of sleep remains elusive, there are many hypotheses that include the restoration of the organism’s energy and the promotion of memory consolidation, which is necessary for acquiring knowledge during wakefulness [7,8]. Additionally, during sleep, the brain is able to eliminate toxic waste products, such as beta amyloids, tau aggregates, and others, via the glymphatic system [9,10,11].

It is generally accepted that sleep secures the maintenance of homeostasis, supports synaptic plasticity and, in due course, memory consolidation [12,13]. The brain states during sleep involved in memory consolidation have recently been reviewed by Born and collaborators [13].

Different brain regions, including the neocortex, hippocampus, thalamus, and brainstem, are involved in the neuronal activities during the various sleep states [14]. However, the neuronal activities underlying the NREM and REM stages are not always separable; different brain centres are involved in both REM and NREM stages [15,16]. Moreover, occasionally, slow-wave activity (associated with the NREM stage) can be detected while REM sleep is still on going [17,18].

How sleep states are regulated remains a mystery, although recent discoveries have begun to unravel some of the molecular, cellular, and neural circuit processes that are involved [19,20]. Historically, the control of sleep has been largely attributed to the interaction of hypothalamic and brainstem nuclei [21] and, recently, this has been extended to include densely distributed networks spanning the forebrain, midbrain, and hindbrain [22].

The brain activity during sleep is typified by circuit-specific oscillations, consisting of slow waves, spindles, and theta waves, with the anatomical substrate based on the thalamocortical or hippocampal networks. For more than 70 years, following the detection of the ascending reticular activating system, our understanding of the various neural circuits that control sleep states has been broadened. Investigations based on pharmacological treatments, various types of stimulation, and lesions have detected specific important sleep-promoting brain regions. These studies have raised the question if specific neurons in those brain areas are responsible for initiating and maintaining the different sleep states [23].

The balance between excitatory glutamatergic and inhibitory GABAergic neurons in the cortico-thalamo-cortical circuitries is essential to sustaining the predominant slow oscillations (<1 Hz), delta waves (1–4 Hz) and sleep spindles that occur during NREM sleep (see Figure 1). In addition to the function of the GABAergic cells in this network, there has been a recent identification of glutamatergic sleep-promoting neurons in other brain regions, such as the CALCA-expressing glutamatergic perioculomotor neurons projecting to the preoptic area (POA) and the medulla [24]. Sleep-promoting glutamatergic neurons have also been identified in the ventrolateral periaqueductal gray (vLPAG) of the midbrain, and in the posterior thalamus [25]. A recent study has uncovered the control of NREM sleep by ventrolateral glutamatergic neurons that project to the preoptic area [26,27].

Glutamate is the main excitatory neurotransmitter that globally regulates sleep and wake states via different glutamate receptors [28]. One receptor of particular interest is the metabotropic glutamate 5 (mGlu5) receptor, which is found expressed in the areas that are involved in the control of sleep states, such as the neocortex, thalamus, and brainstem, where it modulates synaptic transmission, rather than mediating transmission (Figure 1). There is gathering evidence for the involvement of mGlu5 receptors in physiological sleep states, and in pathological states such as spike and waves discharges (SWDs). SWDs are characteristics of various types of generalised epilepsies, as observed in cortical and thalamic EEG recordings. These SWDs are initiated in the cortex and require an intact cortico-thalamo-cortical network [29,30,31].

In the last decade, compounds that allosterically modulate mGlu5 receptors have been developed as novel therapeutic agents, since this receptor can activate an array of signalling pathways (Figure 1). These can be either positive allosteric modulators (PAMs) that, in theory, preferentially activate desirable intracellular signalling pathways while avoiding, or indeed blocking, those pathways that lead to adverse or unwanted effects of orthosteric agonists; in contrast, negative allosteric modulators (NAMs) inhibit mGlu5 receptors, regardless of the amount of orthosteric activation [32]. Some of these molecules have been used in preclinical and human positron emission tomography (PET) studies [33,34]. In the rest of this review, we will focus on the mGlu5 receptors. The role of the mGlu receptor subtypes in sleep cycles has been recently reviewed by Holter and collaborators [35].

## 2. Activity in the Thalamocortical System Shaping Physiological and Pathophysiological Oscillations

The correct regulation of rhythmicity within the cortico-thalamo-cortical network of the brain is required in various aspects of sensation and cognition, as well as for the regulation of wakefulness and sleep, while thalamocortical dysrhythmias have been suggested to underlie a multitude of neurological and psychiatric disorders, including epilepsy, schizophrenia, attentional problems, and mood disorders [36].

During wakefulness, glutamatergic thalamocortical relay neurons have a membrane potential of about −60 mV. In this state, environmental stimuli evoke a tonic firing pattern in thalamic relay and cortical pyramidal neurons, shaping the high frequency, low amplitude surface EEG, characteristic of wakefulness and full conscious perception [37] (Figure 1). Upon falling asleep, the inhibitory GABAergic input of the reticular thalamic nucleus (nRT) gradually hyperpolarizes thalamocortical neurons, switching them into a burst firing pattern [38,39]. Associated with this switch in firing pattern is the occurrence of rhythmic alpha and theta waves as well as sleep spindle oscillations, recorded in the surface EEG during drowsiness and light sleep, as well as the occurrence of slow delta waves during deep sleep, which is consistent with a reduction in the sensory information that is relayed to the cortex [40,41] (see also Figure 1). A prototypical thalamocortical dysrhythmia, in which rhythmic hypersynchronous burst fining is observed in cortical and thalamic neurons, is childhood absence epilepsy (CAE), which is found in about 10% of children with epilepsy. Patients experience up to several hundred seizures per day, characterized by reduced consciousness and concomitant synchronous, bilateral spike-wave discharges (SWD), as can be seen in the EEG [42,43] (see also Figure 1).

Thalamic relay (so-called first-order) neurons pass sensory information in a precise topographic manner to either the primary auditory, visual or somatosensory cortex; higher-order thalamic nuclei receive their driving input via the glutamatergic cortico-thalamic neurons [44]. Higher-order thalamic nuclei are proposed to support long range cortico-cortical communication and produce generalization of activity [45,46] (see also Figure 1). Interestingly, network analysis of local field potential recordings acquired in the somatosensory cortex, nRT, higher- and first-order thalamic nuclei of freely behaving rats, revealed that higher-order nuclei showed different changes in coupling during the process of falling asleep, and a deepening towards slow wave sleep, compared to first-order thalamic nuclei [47]. The latter might be partially explained by a different projection pattern of the nRT to first- versus higher-order nuclei, enabling the nRT to hyperpolarize thalamic nuclei differentially [48,49]. As a consequence, the nRT has been associated with the occurrence of local sleep [50]. Local sleep phenomena, such as the locally restricted nesting of sleep spindles into the upstate of slow delta oscillation, are thought to be essential for memory consolidation [51]. In patients and animal models of schizophrenia, a reduction in the number of sleep spindles has been described. Enhancement of spindle activity by eszoplicone, however, was not enough to improve memory consolidation in schizophrenia, but required a correct nesting of spindles and slow oscillations [52,53].

In general, the generation of rhythmic activity, and the regulation of neuronal excitability, in the thalamocortical system is thought to predominantly rely on the interplay of three ionic currents governed by calcium (CaV3), HCN, and potassium channels, which are differentially expressed across cortical and thalamic neurons [54,55,56].

Likewise, mGlu5 receptors are differentially expressed throughout the thalamocortical system. The subsequent sections of this review will explore the potential of these receptors in the modulation of physiological and pathological oscillations that are generated in the above-mentioned network (see also Figure 2).

## 3. The Role of Glutamate in Sleep and Wakefulness

The extracellular level of glutamate, measured in the rat orbitofrontal cortex, varies across vigilance states, being minimal during non-rapid eye movement (NREM) sleep, and higher during wakefulness and rapid eye movement (REM) sleep [57]. Similarly, the concentration of glutamate in the motor and prefrontal cortex in freely moving rats increased during waking and REM sleep and decreased during NREM sleep [58]. Earlier studies showed sleep-dependent changes in the rostromedial medulla [59]. Since the action of mGlu receptors is dependent on the amount of extracellular glutamate, this already suggests that the role of mGlu receptors may vary dependent on the state of vigilance, or it is different in the wake period versus the sleep period, and its actions may differ depending on the circadian phase [59,60]. Other evidence for the role of glutamate in sleep and wakefulness comes from pharmacological studies, which showed that microinjections of glutamate during wakefulness into the pedunculopontine tegmentum (PPT), a brainstem structure that is thought to be of key importance for the generation of REM sleep, induced REM sleep [61], while NMDA receptor antagonists increased NREM delta waves [62].

The main support for this arousal system is the glutamatergic input from the parabrachial nucleus and pedunculopontine tegmental nucleus to the basal forebrain, and the GABAergic and cholinergic neurons in the basal forebrain, from which the cerebral cortex is innervated. Disruptions occurring in these areas result in a complete loss of consciousness, while lacerations of either supramammillary dopaminergic or glutamatergic neurons in the ventral periaqueductal gray matter near the dorsal raphe nucleus result in an almost 1/5th loss of the time spent in wakefulness [63].

There has been a hunt for sleep-promoting cells and brain regions, stepping out of those traditional areas in the brain stem and hypothalamus. New glutamate excitatory sleep regulating cells and circuits have recently been identified, such as glutamatergic projections onto POA which promote sleep, and also the identification of subpopulation of glutamatergic NREM sleep inducing neurons in the posterior thalamus [19]. These indications point to the involvement of glutamatergic system in wakefulness.

## 4. mGlu5 Receptors Expression in Sleep-Promoting Brain Regions: Cortico-Thalamo-Cortical Network?

Following the discovery of the ascending reticular activating system more than 50 years ago [64], investigations into sleep-promoting neural circuits has been very limited [21,65].

As previously mentioned, the interplay between excitation and inhibition in the cortico-thalamo-cortical network is relevant in maintaining slow oscillations (<1 Hz), delta waves, and spindles that occur during sleep. It is worth exploring how the presence of mGlu5 receptors in this network modulates these oscillations (see also Figure 2).

mGlu5 receptor subtypes are expressed in the cortex and thalamus, the brain regions in which slow oscillations, delta waves and sleep spindles are generated. They are distributed at relevant synapses within the cortico-thalamo-cortical network (Figure 2). In the rodent neocortex, mGlu5 receptors are significantly expressed postsynaptically on the pyramidal neurons receiving inputs from thalamocortical neurons [66], and there is evidence that these receptors are located in presynaptic terminals [67]. They are also found in numerous regular spiking inhibitory interneurons [68]. The expression pattern of the mGlu5 receptor in the neocortex versus thalamus is in the main complement to that of the mGlu1 receptor expression [69].

Thalamo-cortical neurons show moderate immunostaining of mGlu5 receptors on their dendrites postsynaptic to presynaptic cortical inputs [67,70,71]. The nRT neurons show moderate to low mGlu5 mRNA and protein expression [67,72]. In human post-mortem tissues, mGlu5 receptor is expressed in radially arranged cellular processes, in the neuropil and neocortical neuronal cell bodies. This is in accordance with the pattern of expression found in rodents [71,73]. Therefore, the mGlu5 receptor may contribute to the sleep/wake processes that involve the cortico-thalamo-cortical network (see also Figure 2).

## 5. The Regulation of Sleep States by mGlu5 Receptors

The mGlu5 receptors that = are abundantly expressed in the thalamocortical network (reviewed in [30]) are known to be involved in the generation of the various sleep–wake oscillations, including slow and delta oscillations, and the sleep spindles (see Figure 1 and Figure 2) [74].

Both the slow and delta oscillations arise from variations in the resting membrane potentials of both thalamic and cortical neurons [74,75]. These reciprocally interconnected cells can switch between depolarized, the so called ‘UP’ state, and hyperpolarized, the ‘DOWN’ state. High neural activity occurs during the UP state, and low activity during the DOWN state, and this dynamic interplay is responsible for the slow oscillations. The part of the thalamus that generates the classical waxing and waning of sleep spindles is the nRT [74]. The amount of hyperpolarization is crucial for whether sleep spindles or delta oscillations are produced by the TC cells, and their GABAergic input from the nRT (see Figure 1 and Figure 2).

In the last decade, there have been enormous efforts in developing specific mGlu5 receptor allosteric modulators, positive and negative allosteric modulators (PAMs and NAMs), which bind to sites found in the 7-TM domain of the receptor. Allosteric ligands of the mGlu5 receptor remain an attractive therapeutic strategy for different neurological and psychiatric disorders and comorbid sleep disturbances [75].

Different research groups have investigated the effects of mGlu5 receptor ligands in different animal models. For example, the effects of mGlu5 receptor PAMs on sleep oscillations in rats were investigated in a 20 h continuous EEG study. The drugs were administered at the beginning of the light period [76]. This study showed that the two mGlu5 receptor PAMs (ADX47273 and LSN2814617) have wakefulness-promoting properties.

In earlier studies, the wakefulness-promoting effects of different mGlu5 receptors PAMs (CDPPB, VU0364289, DPFE, LSN2463359, LSN2814617, ADX47273) occur at the cost of (deep) NREM and REM sleep [76,77,78,79]. In general, this variety of mGlu5 receptor ligands produces dose-dependent increases in wakefulness and sleep onset latency; this was accompanied by decreases in NREM and REM sleep. However, not all of these studies distinguished between light and deep NREM sleep or delta power, and active and passive wakefulness.

The first indication for a role of the mGlu5 receptor in the generation of delta oscillations in the thalamus comes from a study in which systemic administration of the NAM, MPEP, increased the power of the low slow wave component (<2 Hz) of the thalamic EEG recorded in vivo in anesthetized rats [80]. Ahnaou et al. (2015) [81] assessed the effects of two mGlu5 receptor NAMs and distinguished between light and deep NREM sleep. They established that the two NAMs (MTEP and MPEP) promoted deep NREM sleep, and lacked behavioural-activating effects, thus did the opposite of the PAMs. NAMs also decreased REM sleep, like what has been observed for PAMs. They also reported an increase in the connectivity (coherence) in the 4–6.5 Hz band in the parietal and occipital cortex, and an increase in the 32–48 Hz band throughout the whole cortex for the NAM MTEP. The increase in deep NREM sleep is striking, considering that rats already have a high amount of deep NREM sleep during the beginning of their habitual sleep period. In summary, it appears that blocking the mGlu5 receptor enhances sleep and facilitates cortical network connectivity, both of which play an essential role in neural plasticity and cognition.

The dose-dependent decrease in REM sleep found by Ahnaou et al. (2015) [81] immediately after the administration of both NAMs and PAMs remain puzzling. In both cases, the effects on REM sleep were accompanied by a dose-dependent decrease in the intermediate stage (IS). IS, is most apparent in the sleeping EEG of rats and cats, is a short-lasting stage which occurs just prior to REM sleep and is characterized by high amplitude frontal sleep spindles and low frequency high amplitude theta activity in the hippocampus, with the lowest thalamic signal transmission of all sleep–wake states. Only in the case of NAMs there is a decrease in both IS and REM sleep, followed by a compensatory increase in both IS and REM sleep [81]. This finding suggests that the stage of sleep allowing REM sleep to occur, IS, needs a delicate, balanced amount of glutamatergic neurotransmission for its occurrence, and to allow a permissive and normal amount of REM sleep. Thus, the key to REM sleep occurrence is either the correct amount of glutamate signalling or an optimal glutamate–GABA balance. The finding that a delayed rebound in IS and REM occurred only after the administration of both NAMs shows that the homeostatic control of REM sleep is disrupted by an excessive amount of glutamatergic neurotransmission, which occurs after the administration of the PAMs [81].

Holter et al. (2021) [75] showed a decrease in REM sleep in response to the administration of the mGlu receptor NAM VU0424238 to rats. The compound, administered two hours after light onset, produced dose-dependent increases in wakefulness during the first four hours of the light cycle, followed by reductions in wakefulness in the dark cycle. This NAM decreased the duration of non-REM sleep immediately after its administration, and this was followed by an increase with the highest dose in the subsequent dark phase of the 12 h to 12 h light–dark (L–D) cycle. It is, however, not clear whether this late reaction is due to a homeostatic or a circadian factor. Holter et al. (2021) [75] also showed that the NAM VU0424238 dose-and-time dependently increased the delta power during NREM sleep, with more than 20% lasting up to ten hours for the highest dose [75]. Delta power is a biomarker for the qualitative aspects of NREM sleep and, although NREM sleep duration was initially reduced, the quality of the delta power is firmly dose- and time-dependently enhanced. There are currently not many hypnotic drugs that have this property [75].

## 6. Sleep in mGlu5 Receptor (^−/−^) (Knockout) Mice

A new element was added in sleep studies regarding the role of mGlu5 receptors: constitutive knockout mice and sleep deprivation. After a baseline period, knockout mice were deprived of sleep for 6 or 8 h, and the recovery of sleep was analysed. Ahnaou et al., in (2015) [81] showed that, in the base-line period, the duration of NREM sleep was slightly enhanced with an accompanying decrease in 1–2 Hz activity during NREM sleep, and an increase in gamma oscillations during wakefulness, while REM sleep was reduced compared to wildtype (WT) littermates. Next, the number of transitions from NREM to REM (intermediate state) and REM to NREM was lower, particularly in the dark period. The recovery of their sleep following deprivation was characterized by the expected increased NREM sleep delta power; however, the increase was lower and shorter-lived in the mGluR5 (^−/−^) receptor mice compared to the WT littermates. The knockout mice did not exhibit the usual recovery of REM sleep, suggesting that the homeostatic drive for REM sleep was lacking. These fascinating results regarding the role of the mGlu5 receptor and, in particular, its ability to increase NREM sleep duration accompanied by a decrease in delta power, suggest that the build-up of the delta during the base-line period is diminished, and this hypothesis is confirmed by the reduction in delta activity in the recovery sleep. This suggests that mGlu5 receptor activation is necessary for high-quality NREM sleep following a period of wakefulness [81].

Aguilar and collaborators [82] confirmed the increased time in NREM sleep, and decreased time awake, in mGluR5 (^−/−^) receptor mice across the base-line period compared to WT (C57BL6) mice, and this was reversed in the last few hours of the dark period. The commonly reported reduction in REM sleep was not observed. Knockout mice demonstrate a more fragmented sleep and wake pattern, as well as shorter NREM episodes, during the dark period. There were no differences observed between the knockout and WT mice in the recovery sleep after 6 h of sleep deprivation, except that the timing of the rebound sleep was different, and occurred earlier, in knockout mice compared to wildtype [81].

Aguilar and collaborators [82], showed a clear difference across the first six hours of the dark phase of the 12-12 light-dark (LD) cycle regarding the dynamics of delta (0.75–4.0 Hz) power during NREM sleep. The regular build-up of a sleep need was greatly attenuated in the knockout mice compared to control groups, although the amount of NREM sleep did not differ significantly. This reduced delta power during the dark period could also mean that mGlu5 receptor controlling sleep and wake have a distinct role in the light and dark period, or it is different during the main sleep and active period, as was recently suggested [81]. Holst and collaborators [31] emphasized the reduced delta power at the beginning of the recovery period in knockout animals. The mGlu5 receptor exerts its influence on delta power during the dark period, suggesting that the mGlu5 receptor exerts a different function during the dark phase versus the light phase. The fact that many other sleep-related changes were found to be different in the light and dark period supports this view. The repeatedly reported decreases in gamma oscillations in knockout mice, as well as after the administration of various NAMs, might be related to how the mGlu5 receptor promotes concentration and alertness, the increased availability of mGlu5 receptors could lead to disinhibition of the gamma oscillations [83].

## 7. The Role of mGlu5 Receptor in Sleep in Humans

Fenobam, a potent selective non-benzodiazepine anxiolytic that acts as a mGlu5 receptor NAM, was found to favour wakefulness over the placebo group, with significant differences in sleep architecture observed between the groups. Specifically, the Fenobam group preferred wakefulness and NREM stage 1 sleep over NREM stages 2 and 3 and REM sleep [84]. The Fenobam group also had an overall decrease in sleep time, with increased delta power and decreased amplitude of sleep spindles, and a reduction in power of alpha, beta, and theta activity during REM sleep. Interestingly, these results contrast somewhat with animal studies, such as those by Ahnaou et al. (2015) [76] where the results regarding the duration of sleep phases were opposite. However, a fairer comparison would be to evaluate Fenobam in a rat sleep study.

There is evidence that expression of the mGlu5 receptor is increased in sleep-deprived patients correlating with an increased need for sleep: 33 or 40 h of continuous wakefulness increased mGlu5 receptor availability, as measured by positron emission tomography or magnetic resonance spectroscopy, and this correlated positively with behavioural and EEG-based biomarkers of an elevated sleep requirement [31,85]. Thus, those subjects who were most affected by sleep deprivation exhibited the largest increase in mGlu5 receptor binding in multiple brain regions, such as the anterior cingulate cortex, insula, medial temporal lobe, parahippocampal gyrus, striatum, and amygdala [85].

## 8. Sleep and Epileptic Oscillations in the Thalamo-Cortical Network

In humans, sleep starts from stage 1 in NREM sleep (N1), then progresses to N2, N3 and REM sleep [86]. The cortical EEG recordings show rhythmic features with lower frequency and larger amplitudes as NREM goes deeper (also see Figure 1). The nRT is the part of the thalamus that generates the classical waxing and waning of sleep spindles, which are brief (0.5–3 s) oscillatory events in the 9–14 Hz range. The NREM sleep observed in rodents is considered to be just one stage, and it alternates with the REM sleep stage in a fragmented manner [87]. People with either acquired or genetic epilepsies usually experience seizure activity during sleep which, in turn, will affect the sleep structure. This nocturnal sleep fragmentation leads to cognitive problems in people with epilepsy, which might persist even after the seizures are no longer occurring [88]. This may lead to reduced quality of life and higher mortality. In genetic epileptic animals of the absence type, the typical SWDs preferentially occur during drowsiness and at unstable vigilance periods [89]. However, the occurrence of SWDs could not be predicted from the preceding brain state by more than 2 s [90]. There is also evidence that neuroplastic changes occur within the networks in which the epileptic activity is generated and expressed, but that other networks are equally affected [91]. In the last decade studies are beginning to characterise the main mechanisms of the starting and stopping of seizures that occur in absence epilepsy [46]. Our present knowledge of the cellular and circuit mechanisms points to the involvement of the cortico-thalamo-cortical network in the initiation, maintenance and stopping of the aberrant EEG activity [92,93,94].

There are a large number of studies demonstrating that SWDs, as seen in rodent absence models, result from malfunctioning of the cortico-thalamo-cortical network [95]. As this network is involved in slow-wave sleep oscillations, such as delta waves and sleep spindles that are important for physiological sleep, it implies that the aberrant cortico-thalamo-cortical SWDs may result in sleep disorders [96,97,98]. SWDs in WAG/Rij rats are associated with a slight non-REM sleep disorder, next to a firm reduction in the number of REM sleep periods, and total REM time sleep and a prolonged intermediate stage [95].

The WAG/Rij rat strain is a polygenic model that evolves different genes impacting the epileptic phenotype [99]. These rats show bilateral spike and wave discharges (SWDs) on the electrocorticography (ECoG) recordings (see also Figure 1), which are developed between two and four months of age. The SWDs appear spontaneously, with a duration of 1–15 s, and are characterized by a frequency of 7–9 Hz. This epileptic activity and behaviour mimic childhood absence epilepsy, and the response to ethosuximide, by reducing the SWDs. Other antiepileptic drugs, such as carbamazepine, or those that mimic the action of GABA, such as vigabatrin and tiagabine, increase SWDs if acutely administered, which is also the case in humans with absence epilepsy. Studies in the WAG/Rij rat models have shown that targeting the mGlu5 receptors is an important strategy for the development of drugs for absence epilepsy, since the mGlu5 receptor PAMs used are efficacious in reducing SWDs and had anti epileptogenic effects [100,101,102,103]. The PAMs did not behaviourally activate the animals. However, it is intriguing that the used PAM (VU0360172) showed opposite interaction effects with the GAT1 blocker tiagabine, when they were locally co-infused into the thalamus or cortex [104]. SWDs were reduced by co-cortical infusion and enhanced when co-administered in the dorsal thalamic area. Therefore, the effects of the mGlu5 receptor activators might differ in different brain regions through their different interaction with GABAergic neurons. Another consequence of this study, and the in vitro data as reported by Celli et al. [105], was that the EEG does not tell the whole story about what is going on subcortically regarding the consequences of mGlu5 receptor activation, and their interaction with GABA. An intriguing issue remaining is that REM sleep is suppressed by both mGlu5 receptor activation and inhibition. The pharmacological data obtained regarding IS points towards a crucial role for this sleep stage in controlling the number and duration of REM sleep episodes. Additionally, the changes in IS characteristics of WAG/Rij were accompanied by fewer REM sleep periods. Traditionally, REM sleep onset is the result of increased activity of cholinergic cells of the peri-brachial nuclei of the mesopontine brain stem, and the offset by the co-activation of noradrenergic and serotonergic neurons in the locus coeruleus and dorsal raphe. However, there is also evidence that the genesis of REM sleep is due to the interaction of GABA and glutamate in these brainstem nuclei [106] and, in this way, the effects of mGlu5 receptor modulation on REM sleep and IS might be partly explained.

Whether there is a shift in the sleep structure of these epileptic rats against non-epileptic control rats following treatment with mGlu5 receptor PAMs or NAMs is a question that warrants future investigation. Additionally, there is evidence from sleep studies that mGlu5 receptor modulation may have different effects in the main wake and main sleep period and are differently expressed in the light versus dark periods [107]. It is therefore suggested that subsequent sleep–wake studies, or studies regarding the anti-absence action of mGlu5 receptor modulators are carried out in the dark and light period of the 24 h L–D cycle.

## 9. Conclusions

It is evident that there is an intimate relationship between sleep and epilepsy from preclinical and human studies. Sleep architecture is fragmented in epilepsy, and antiepileptic medications may also reduce sleep quality. Seizures may preferentially, but not exclusively, occur during N1 and N2 sleep, or can be triggered by arousals during sleep. Interictal discharges are also seen more commonly during sleep, while sleep also may modify the electroclinical morphology and seizure distribution.

In contrast, during REM sleep, seizure occurrences are mostly reduced as compared to wake states [87,108].

Preclinical studies from WAG/Rij rats pathological oscillation SWDs, as shown on ECoG recordings, have allowed us to get a better understanding of how mGlu5 receptors tune the thalamo-cortical circuity [30,109]. For instance, Crunelli and collaborators have demonstrated that pathological SWDs from different rodent models of absence epilepsy are driven by enhanced tonic GABA currents in the thalamo-cortical neurons, which is a result of a reduced GABA reuptake due to a defective GABA transporter 1 in the thalamus [110]. Interestingly, a drug that tonically inhibits thalamo-cortical cells and is a GAT transporter blocker, tiagabine, not only enhances SWDs, but is also a sleep-promoting drug. This increased both NREM sleep and the low frequency (delta) activity in the EEG within non-REM sleep [111]; it also delays the onset of REM sleep in healthy rats [112]. Therefore, we can suggest that tonic inhibition in the thalamus might be responsible for the occurrence of SWDs, and that tonic inhibition in brain-stem regions for the diminishment of REM sleep in the first two hours after administration. In a recent study, we have shown that the mGlu5 receptor PAM, VU0360172, increases the expression of the GAT-1 protein and GABA reuptake in the thalamus [105]. This could be one of the key mechanisms by which the mGlu5 receptor modulates thalamo-cortical networks. Hypothetically, the thalamo-cortical events are primarily based on the interaction between different ionic currents driven by the calcium (CaV3), HCN, and potassium channels. Futures studies, testing the crosstalk between the mGlu5 receptor and these ion channels in vivo in animal models (that mimic human epilepsy types), would be required to shed light on cellular and circuit mechanisms that are involved in sleep oscillations, and the source of pathological oscillations that occur during sleep.

Among the different types of sleep disorders, insomnia is the most common, characterized by a difficulty in initiating or maintaining sleep, early morning awakening, or non-restorative sleep. Insomnia can have significant negative effects on the quality of life and can lead to a range of physical and mental health issues [113]. There is evidence to suggest that the targeting of mGlu5 receptors could be a potential strategy for treating sleep disorders. mGlu5 receptors are primarily located in brain areas involved in sleep–wake regulation and expression of the slow oscillations. They play a role in regulating several physiological processes, including NREM sleep, e.g., via targeting thalamo-cortical cells. Studies have shown that mGlu5 receptor antagonists and NAMs can increase NREM sleep and, in particular, slow wave activity during NREM sleep, and decrease wakefulness in animal models of insomnia [74]. Moreover, they participate in the normal homeostasis of REM sleep after an initial decrease. The studies using mGlu5 knockout mice revealed a deficiency in the build-up of the sleep drive [114], indicating that the mGlu5 receptor has a role in this process [115], which is important for healthy sleep with sufficient delta activity.

However, it is important to note that further research is required to fully understand the potential of targeting mGlu5 receptors for the treatment of sleep disorders in humans. While these initial findings are promising, more fundamental research and clinical trials are necessary to confirm the efficacy and safety of mGlu5 receptor modulators in the treatment of sleep disorders.

## Figures and Tables

**Figure 1 cells-12-01761-f001:**
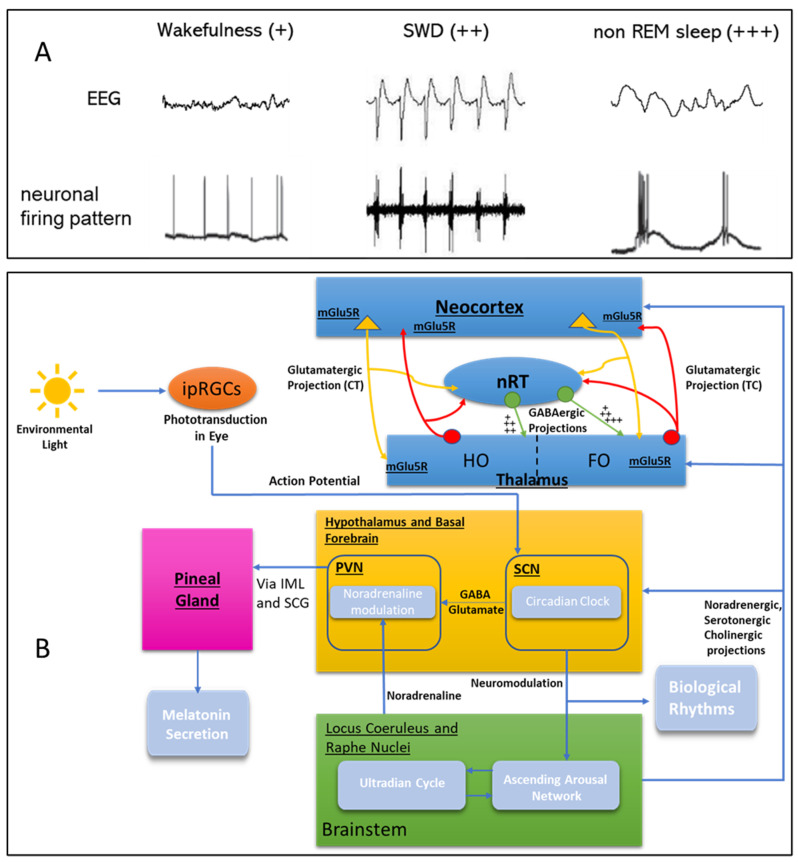
A brief overview of the control of the sleep–wake systems, illustrating the involvement of glutamate. Diagrammatic representation of the general regulatory circuits and brain regions involved in the sleep–wake and ultradian cycles, as well as their major inputs and outputs. Elements of this network are discussed in further depth in the review. Top panel (**A**) depicts changes in the firing mode of thalamo-cortical neurons from tonic to burst firing, and their correlated pattern in the surface EEG seen during wakefulness, SWD, and NREM-sleep. Note that the change is caused by a gradual increase in inhibition provided by the nRT, resulting in a more hyperpolarized membrane potential of thalamocortical neurons. Different degrees of inhibition are indicated as + (low), ++ (medium), and +++ (strong), respectively. Abbreviations: IML, Intermediolateral Nucleus; ipRGC, intrinsically photosensitive retinal ganglion cell; mGlu5R (=mGlu5 receptor), metabotropic glutamate receptor 5; nRT, thalamic reticular nucleus; PVN, paraventricular nucleus of the hypothalamus; SCG, superior cervical ganglion; SCN, suprachiasmatic nucleus, SWD spike and wave discharge, HO higher-order thalamic nuclei, FO first-order thalamic nuclei (**B**).

**Figure 2 cells-12-01761-f002:**
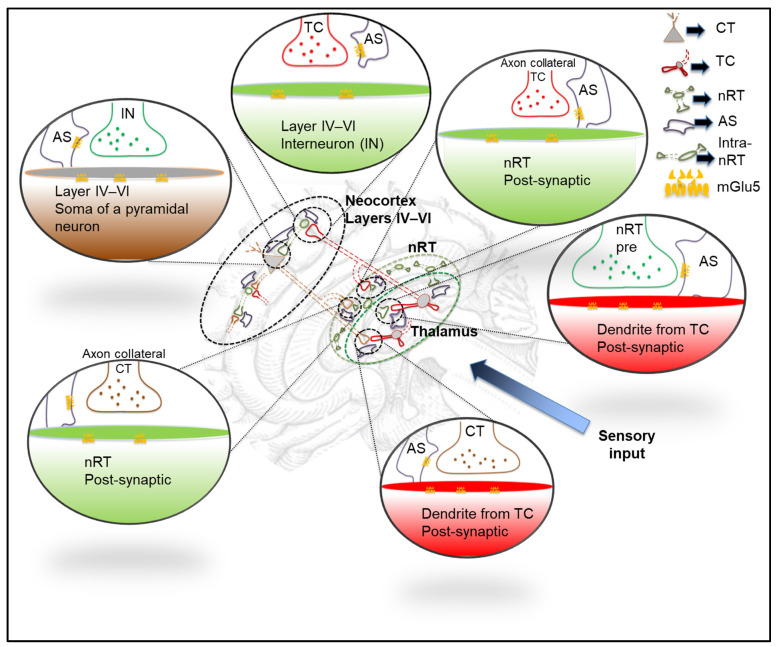
Basic features of the thalamocortical and cortico-thalamo-cortical network underlining the synaptic localization of mGlu5 receptor in yellow. In brief: The thalamic nuclei receive and process sensory inputs from the periphery, while the higher-order receives and processes inputs from layer V in the neocortex. The sensory afferents to the thalamic nuclei are represented by the dark graduated arrow (sensory input). The glutamatergic thalamo-cortical neurons (TC) from the rest of the thalamus (in red) are projecting to the layer V interneuron (IN) and the pyramidal neurons (CT), represented as a triangle in grey and axons as brown. In turn, the pyramidal neurons (CT) send their glutamatergic projections back to the thalamic neurons, and collaterals to the reticular thalamic neurons (nRT), which are all GABAergic. Note that the TC neurons (in red), as they project to the neocortex, send collaterals onto the nRT as well. Note that the subcellular distribution of the mGlu5 receptors, which are mostly located perisynaptically at asymmetrical synapses (post-synaptic), are depicted in yellow. The axon terminals in most of the synapses are ensheathed by the astrocyte processes, depicted in purple. mGlu5 receptors are also present in the astrocytes (AS).

## Data Availability

Not applicable.

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
