# Peer review of "The Metabotropic Glutamate 5 Receptor in Sleep and Wakefulness: Focus on the Cortico-Thalamo-Cortical Oscillations"

_cells, 2023, doi:10.3390/cells12131761_

Round 1
Reviewer 1 Report
This is a well thought out and written review on the involvement of mGlu5 receptors in the sleep-wake process. The authors did this on the basis of well-chosen literature of recent years. A particular advantage of the work, which I emphasize, is the description and discussion of the role of the described receptor in relation to sleep pathology, especially insomnia and epilepsy. Especially that the corresponding author is one of the leading researchers of the WAG/Rij rat strain, in whom the connection between spike and wave discharges (SWDs) and sleep rhythm disorders has been proven. Further research on the involvement of mGlu5 receptors in sleep pathology may lead to therapeutic effects, which the authors of the paper point out.
The second sentence of the Introduction requires comment. I'm not sure if drospohil's torpor can be called sleep, in its classical definition that sleep is a "special state of the brain"? Before defining sleep in different groups of animals, I think it is necessary to give a definition of sleep.
Author Response
Author’s Response to Reviewer 1
We thank the reviewer for this useful comment:
We have amended the second sentence of the introduction and given a general definition of sleep and commented that sleep from a behavioral point of view can be observed across various species (ranging from mammals to invertebrates such as drosophila).
Hopefully, this amendment is satisfactory.
Reviewer 2 Report
Overall, this is an interesting review that considers how mGlu5 receptors may be involved in regulation of sleep/wake and whether targeting such receptors may be useful in sleep disorders/epilepsy
The review begins with a general introduction to sleep phases and the cortico-thalamic circuitry and glutamatergic/GABAergic regulation of such circuitry.
Comments/corrections
There are lots of typos and various grammatical errors that need to be corrected – far too many for me to list.
There are various aspects that need much fuller explanation – for example, P2:
‘The metabotropic glutamate 5 receptor subtype is potentially well positioned in different brain areas to modulate synaptic transmission rather than mediating [fig 1].
‘In the last decade compounds that allosterically modulate mGlu5 receptors have been developed as novel therapeutic agents as this receptor has an array of signalling endpoints [fig.1]’.
Both of the above statements refer to Fig 1 but it is not clear how those statements relate meaningfully to the information in Fig 1.
Figure 1 needs to be better integrated into the review – as an example section 2 refers to oscillations across the thalamocortical circuit – this type of information could have been usefully included into Fig 1.
Section 3 (para 2) also refers to Fig 1 but it is not clear how the reference to GABA/glu and slow oscilations,delta waves and spindles is explained in Fig 1.
Figure 2 is conveying a lot of information and takes a lot of effort to understand. The abbreviation AS is not explained – I assume it stands for astrocyte? The yellow mGlu5Rs don’t stand out well on the different colours in each of the different images. In the text it states that mGlu5Rs are also found presynaptically, but this isn’t shown at any of the synapses depicted in Fig 2.
P6: Does the following refer to a figure in this review? If so I couldn’t find anything that relates to the statement. ‘The backbone of the arousal system in this model, shown in red, is the glutamatergic input from the parabrachial nucleus (PB) and pedunculopontine tegmental nucleus (PPT) to the basal forebrain, and the GABAergic and cholinergic neurons in the basal forebrain (BF) that diffusely innervate the cerebral cortex.‘
The overall evidence provided for the role of mGlu5 in controlling sleep was patchy and still leaves very many questions – it would have been useful to have had some more critical appraisal of the evidence and what more convincingly still needs to be answered.
The section on sleep disturbances/epilepsy needs to be better fleshed out. It felt underdeveloped but could be a very useful part of this this review if it was more developed.
Author Response
Comments and Suggestions for Authors
Overall, this is an interesting review that considers how mGlu5 receptors may be involved in regulation of sleep/wake and whether targeting such receptors may be useful in sleep disorders/epilepsy
The review begins with a general introduction to sleep phases and the cortico-thalamic circuitry and glutamatergic/GABAergic regulation of such circuitry.
Comments/corrections
- There are lots of typos and various grammatical errors that need to be corrected – far too many for me to list.
Author’s Response to Reviewer:
The authors apologise for the oversight and all related grammatical errors and typos have been corrected.
- There are various aspects that need much fuller explanation – for example, P2:
‘The metabotropic glutamate 5 receptor subtype is potentially well positioned in different brain areas to modulate synaptic transmission rather than mediating [fig 1].
Author’s Response to Reviewer:
This is an important point and we have now made the necessary changes that would clarify this question
- ‘In the last decade compounds that allosterically modulate mGlu5 receptors have been developed as novel therapeutic agents as this receptor has an array of signalling endpoints [fig.1]’.
- Both of the above statements refer to Fig 1 but it is not clear how those statements relate meaningfully to the information in Fig 1.
- Figure 1 needs to be better integrated into the review – as an example section 2 refers to oscillations across the thalamocortical circuit – this type of information could have been usefully included into Fig 1
Author’s Response to Reviewer:
This is an important point and as such has been included in the figures: We have now made the necessary changes that would clarify this question, included an insert to the previous figure 1 and amended the figure legends. We have also corrected the text
- Section 3 (para 2) also refers to Fig 1 but it is not clear how the reference to GABA/glu and slow oscilations,delta waves and spindles is explained in Fig 1.
Author’s Response to Reviewer:
We have now made the necessary changes that would clarify this question, in the text highlighted in yellow
- Figure 2 is conveying a lot of information and takes a lot of effort to understand. The abbreviation AS is not explained – I assume it stands for astrocyte? The yellow mGlu5Rs don’t stand out well on the different colours in each of the different images. In the text it states that mGlu5Rs are also found presynaptically, but this isn’t shown at any of the synapses depicted in Fig 2.
Author’s Response to Reviewer:
This is an important point; the authors apologise for the oversight. We have now increased the resolution of the figure, enhanced the colours, and made changes to the figures as suggested by the reviewer.
- P6: Does the following refer to a figure in this review? If so I couldn’t find anything that relates to the statement. ‘The backbone of the arousal system in this model, shown in red, is the glutamatergic input from the parabrachial nucleus (PB) and pedunculopontine tegmental nucleus (PPT) to the basal forebrain, and the GABAergic and cholinergic neurons in the basal forebrain (BF) that diffusely innervate the cerebral cortex.‘
Author’s Response to Reviewer:
We have rewritten this session and made the necessary changes that would clarify this question, Changes are now highlighted in yellow
- The overall evidence provided for the role of mGlu5 in controlling sleep was patchy and still leaves very many questions – it would have been useful to have had some more critical appraisal of the evidence and what more convincingly still needs to be answered.
Author’s Response to Reviewer:
Thank you for this observation. We have rewritten this session and made the necessary changes that would clarify this question. Changes are now highlighted in yellow
Reviewer 3 Report
This is a concise well-done review of the role of the metabotropic glutamate receptor type 5 in electrical oscillations of the corticothalamo-cortical network during sleep and wakefulness.
Some observations to modify are depicted below:
Major:
1. Describe how authors consider that PAMS and NAMS would be useful in mGluR5 activation in the context of circadian rhythms. Take into consideration the knowledge about the half-life of these components and the chronotherapy as a challenge in the treatment of thalamocortical dysrhythmia in sleep-wake alterations. As well, as consider the extrapyramidal side effects of PAMS and NAMS.
Minor:
1. In the introduction section, reference 5 does not refer to the role of sleep. Several works about this topic could be referenced accurately.
2. I suggest formatting the references according to the authors’ guide.
3. Sometimes, authors refer to metabotropic glutamate receptor 5 as a subtype or as mGlu5 receptors. It would be convenient to be clearer in naming this receptor and if mGluR5 has subtypes (maybe a and b isoforms).
4. Add the correspondent abbreviature the first time that this receptor is mentioned.
5. Describe the abbreviature for SWDs.
6. In line 117 change thamlamic neurons for thalamic neurons.
7. The sentence “mGlu5 receptor might be involved in sleep regulation as these receptors are abundantly expressed in cortex and thalamus, the brain regions in which delta oscillations and sleep spindles are generated” is repetitive in sections 3 and 5.
Author Response
Comments and Suggestions for Authors
This is a concise well-done review of the role of the metabotropic glutamate receptor type 5 in electrical oscillations of the corticothalamo-cortical network during sleep and wakefulness.
Some observations to modify are depicted below:
Major:
- Describe how authors consider that PAMS and NAMS would be useful in mGluR5 activation in the context of circadian rhythms. Take into consideration the knowledge about the half-life of these components and the chronotherapy as a challenge in the treatment of thalamocortical dysrhythmia in sleep-wake alterations. As well, as consider the extrapyramidal side effects of PAMS and NAMS.
Author’s Response to Reviewer:
We have rewritten this session and made the necessary changes that would clarify this question, Changes are now highlighted in yellow
Minor:
- In the introduction section, reference 5 does not refer to the role of sleep. Several works about this topic could be referenced accurately.
Author’s Response to Reviewer:
The reference has been changed accordingly, Changes are now highlighted in yellow
- I suggest formatting the references according to the authors’ guide.
Author’s Response to Reviewer:
The references have been formatted according, Changes are now highlighted in yellow
- Sometimes, authors refer to metabotropic glutamate receptor 5 as a subtype or as mGlu5 receptors. It would be convenient to be clearer in naming this receptor and if mGluR5 has subtypes (maybe a and b isoforms).
Author’s Response to Reviewer:
metabotropic glutamate receptor 5 is now consistently written throughout the text as mGlu5 receptors.
- Add the correspondent abbreviature the first time that this receptor is mentioned.
Author’s Response to Reviewer:
now consistently written throughout the text as mGlu5 receptors
- Describe the abbreviature for SWDs.
Author’s Response to Reviewer:
Spike and wave discharges (SWDs) is now being described in the text
- In line 117 change thamlamic neurons for thalamic neurons.
Author’s Response to Reviewer:
The statement thalamic neurons has now been corrected
- The sentence “mGlu5 receptor might be involved in sleep regulation as these receptors are abundantly expressed in cortex and thalamus, the brain regions in which delta oscillations and sleep spindles are generated” is repetitive in sections 3 and 5.
Author’s Response to Reviewer:
This statement has now been changed and corrected accordingly.